# Suppressing the Heterogeneity: A Strong Feature Extractor for Few-shot Segmentation

**Zhengdong Hu**[1,2][*]**, Yifan Sun**[2]**, Yi Yang**[3] [†]

[1] ReLER, Centre for Artificial Intelligence, University of Technology Sydney, Australia
[2] Baidu Inc.
[3] CCAI, College of Computer Science and Technology, Zhejiang University
huzhengdongcs@gmail.com , sunyf15@tsinghua.org.cn , yangyics@zju.edu.cn

## Abstract

This paper tackles the Few-shot Semantic Segmentation (FSS) task with focus on learning the feature extractor. Somehow the feature extractor has been overlooked by recent state-of-the-art methods, which directly use a deep model pretrained on ImageNet for feature extraction (without further fine-tuning). Under this background, we think the FSS feature extractor deserves exploration and observe the heterogeneity (*i.e.*, the intra-class diversity in the raw images) as a critical challenge hindering the intra-class feature compactness. The heterogeneity has three levels from coarse to fine: 1) Sample-level: the inevitable distribution gap between the support and query images makes them heterogeneous from each other. 2) Region-level: the background in FSS actually contains multiple regions with different semantics. 3) Patch-level: some neighboring patches belonging to a same class may appear quite different from each other. Motivated by these observations, we propose a feature extractor with Multi-level Heterogeneity Suppressing (MuHS). MuHS leverages the attention mechanism in transformer backbone to effectively suppress all these three-level heterogeneity. Concretely, MuHS reinforces the attention / interaction between different samples (query and support), different regions and neighboring patches by constructing cross-sample attention, cross-region interaction and a novel masked image segmentation (inspired by the recent masked image modeling), respectively. We empirically show that 1) MuHS brings consistent improvement for various FSS heads and 2) using a simple linear classification head, MuHS sets new states of the art on multiple FSS datasets, validating the importance of FSS feature learning.

## 1 Introduction

Few-shot semantic segmentation (FSS) aims to generalize the semantic segmentation model from base classes to novel classes, using very few support samples. FSS depicts a potential to reduce the notoriously expensive pixel-wise annotation and has thus drawn great research interest. However, we observe that the current research has been biased towards partial component of the FSS framework. Concretely, an FSS framework typically consists of a feature extractor and a matching head, while the recent state-of-the-art methods (Zhang et al. (2019); Tian et al. (2020b); Li et al. (2021a); Xie et al. (2021b); Wu et al. (2021); Zhang et al. (2021a); Li et al. (2020)) all focus on the matching head. They pay NO effort on learning the feature extractor and adopt a ImageNet-pretrained model without any fine-tuning.

Under this background, we think the FSS feature extractor deserves exploration and take a rethink on the corresponding challenge. Some prior literature (Tian et al. (2020b); Zhang et al. (2021b)) argue that the challenge is mainly because the limited support samples are insufficient for fine-tuning a large feature extractor (*e.g.*, ResNet-50 (He et al. (2016))), therefore leading to the over-fitting problem. We hold a different perspective and observe the heterogeneity (*i.e.*, the intra-class diversity in the raw images) as a critical challenge hindering the intra-class compactness of FSS features. Although the heterogeneity is not a unique problem in FSS (*e.g.*, it does exist in the

---

[*]Zhengdong Hu makes his part of work during internship in Baidu Inc.
[†]Corresponding author.

"cow" → "cattle"   (support → query)

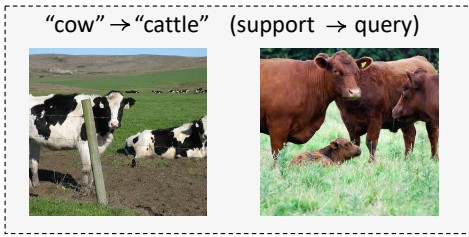

BG: "horse" + "grass"

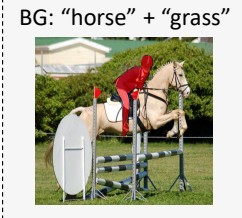

white & dark part of body

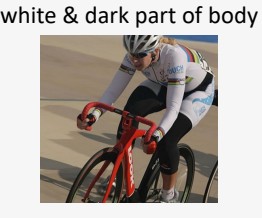

(a) sample-level             (b) region-level           (c) patch-level

Figure 1: The heterogeneity from three levels. (a) shows the sample-level heterogeneity between the support and the query. The "cow" in the support is adopted to segment "cattle" in the query, in spite of their different appearance. (b) shows the region-level heterogeneity in the background. When the foreground object is the "rider", the "horse" should share the same class (BG:background) with "grass". (c) shows the patch-level heterogeneity among neighboring patches. The color of upper and lower part of body is different.

canonical segmentation as well), its challenge is significantly amplified by the few-shot setting. In our viewpoint, the heterogeneity has three levels from coarse to fine:

• *Sample-level* heterogeneity exists between the query and support images due to their distribution gap. For example, in Fig. 1 (a), the foreground objects ("cow" and "cattle") in the support and query images look quite different, although they both belong to a same semantic class "cow".

• *Region-level* heterogeneity exists (mostly) in the background, which actually contains multiple regions with different semantics. In Fig. 1 (b), "horse" in the image is a foreground region when the support object is another horse. However, when the support object shifts to a "rider", the horse in the image should be merged into the background, resulting in the region-level heterogeneity.

• *Patch-level* heterogeneity exists among neighboring patches which belong to a same semantic class but have significant appearance variations. For example, in Fig. 1 (c), the upper and lower body of a single person are in different colors, therefore introducing patch-level heterogeneity.

Motivated by these observations, we propose an FSS feature extractor with Multi-level Heterogeneity Suppressing (MuHS). MuHS adopts the transformer backbone and leverages the attention mechanism to suppress all these three-level heterogeneity. Our choice of using the transformer backbone is natural: the attention mechanism provides strong potential for constructing long-range dependencies across samples, regions and patches. Concretely, MuHS reinforces the attention / interaction between different samples (query and support), different regions and neighboring patches by constructing cross-sample attention, cross-region interaction and a novel masked image segmentation, respectively. To be more specific, these attention / interaction are as below:

(i) *Cross-Sample Attention*. In popular transformers, the attention is within each single sample and does not cross multiple samples. In contrast, MuHS constructs cross-sample attention with a novel design of "linking" tokens. In each transformer layer, we use some linking tokens to connect all the patch tokens from the query and support samples simultaneously, therefore efficiently propagating information across different samples.

(ii) *Cross-Region Interaction*. In popular transformers, the attention usually encourages feature interaction (absorption) between similar patch tokens. In contrary to this common practice, MuHS enforces additional feature absorption between patch tokens from dissimilar regions in the background. Such a cross-region interaction smooths the background and suppresses the region-level heterogeneity.

(iii) *Masked Image Segmentation*. Inspired by the recent masked image modeling (MIM), MuHS randomly masks some patch tokens and makes partial prediction for the existing patches. Afterwards, MuHS fills trainable mask tokens and encourages the deep model to make the holistic prediction for complete patches, yielding a novel masked image segmentation. The learned capacity of inferring the semantics of the masked patches from neighboring patches suppresses the patch-level heterogeneity.

In MuHS, the above three components respectively mitigate a corresponding type of heterogeneity and achieve complementary benefits for few-shot semantic segmentation. Empirically, we show that using MuHS to replace the frozen feature extractor (pretrained on ImageNet) brings consistent improvement for multiple popular FSS heads. Importantly, since the MuHS feature has relatively good intra-class compactness, we simply cooperate it with a linear classification head and achieve new state of the art on multiple FSS datasets. For example, on PASCAL-$5^i$, MuHS achieves 69.1% mIoU under 1-shot setting.

Our main contributions are summarized as follows: First, we shift the FSS research focus from the matching head to the feature extractor and reveal the heterogeneity as an important challenge. Second, we propose Multi-level Heterogeneity Suppressing (MuHS). MuHS utilizes novel cross-sample attention, cross-region interaction and masked image segmentation to suppress the heterogeneity from three levels. Third, we conduct extensive experiments to validate the effectiveness of the proposed MuHS. Experimental results confirm that MuHS is compatible to multiple FSS heads and achieves new state of the art using a simple linear classification head.

## 2 RELATED WORKS

**Few-shot Segmentation Methods With Focus On Matching Head.** The recent state-of-the-art FSS methods (Zhang et al. (2022); Tian et al. (2020b); Li et al. (2021a); Xie et al. (2021b); Wu et al. (2021); Jiao et al. (2022); Lang et al. (2022); He et al. (2023a); Siam et al. (2019); He et al. (2023b)) focus on learning the matching head (based on a frozen CNN feature extractor). Some methods (Zhang et al. (2019); Tian et al. (2020b); Li et al. (2021a); Xie et al. (2021b); Lang et al. (2022); Ding et al. (2018)) generate a prior-map based on the similarity between samples (query and support) and adopt convolution based matching heads to further improve the segmentation accuracy. Zhang et al. (2021b); Lu et al. (2021) proposed transformer-based matching head and perform attention mechanism to aggregate features from support to query. Moreover, some methods (Min et al. (2021); Hong et al. (2021)) propose 4D convolutions to fully extract multi-level features.

Different from these recent progresses, this paper focuses on learning the feature extractor. The proposed MuHS feature extractor brings general improvement for a battery of matching heads and achieves state-of-the-art accuracy with a simple linear classification head.

**Transformers for Visual Recognition.** Recently, transformers are introduced to computer vision tasks, *e.g.*, image classification (Dosovitskiy et al. (2020); Vaswani et al. (2021)), segmentation (Wang et al. (2021); Xie et al. (2021a); Li et al. (2021b; 2022); Zhou et al. (2021)), detection (Carion et al. (2020); Zhu et al. (2020); Bar et al. (2022)) and have shown promising performance.

Under FSS scenario, we observe three-level heterogeneity (*i.e.,* sample-level, region-level, patch-level), which hinders intra-class compactness of FSS features. We think the attention mechanism in the transformer provides strong potential for constructing long-range dependencies across samples, regions and patches. Therefore, the proposed MuHS adopts the transformer network as its backbone and utilizes the characteristics of transformer to suppress all these three-level heterogeneity in a unified framework.

**Masked Image Modeling.** Masked modeling methods (Devlin et al. (2018); Radford et al. (2018; 2019)) are wildly utilized in NLP tasks. BERT (Devlin et al. (2018)) utilizes a "masked language model" (MLM) to randomly masks input and predict the original vocabulary id of the masked tokens. Motivated by BERT, BEIT (Bao et al. (2021)) proposes "masked image modeling" (MIM) to perform self-supervised learning on Vision task. It randomly masks some proportion of image patches and replaces them with a mask tokens. Recently, SimMIM (Xie et al. (2022)), MAE (He et al. (2022)) simplify the MIM designs and improves transformers.

An important component of MuHS is inspired by the masked image modeling to suppress the patch-level heterogeneity. Different from the popular self-supervised scheme, MIS is fully supervised. It masks out some query patches at the input and yet maintains the holistic prediction for segmentation, yielding a novel Masked Image Segmentation (MIS) learning.

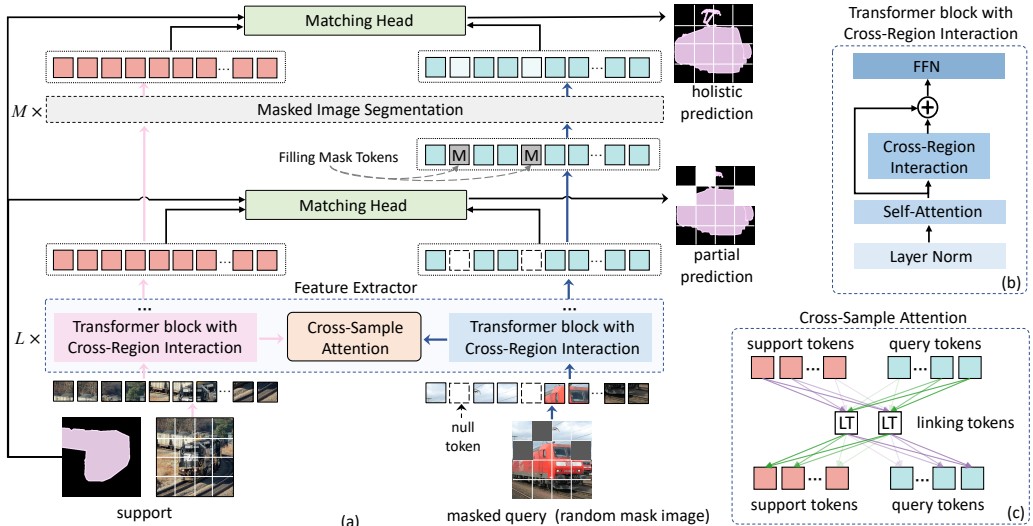

Figure 2: Overview of the proposed MuHS. **(a)** Following the vision transformer backbone, MuHS splits the support and query images into patches, embeds each patch into a patch token and feeds all the patch tokens into the stacked transformer block. Based on the backbone, MuHS 1) inserts a Cross-Sample Attention module between the query and support features to suppress the sample-level heterogeneity (as in Section 3.2), 2) appends an extra Cross-Region Interaction module upon the original self-attention layer to suppress the region-level heterogeneity (as in Section 3.3), and 3) integrates a novel Masked Image Segmentation task to suppress the patch-level heterogeneity (as in Section 3.4). **(b)** The detailed structure of the transformer block with Cross-Region Interaction. **(c)** The detailed structure of the Cross-Sample Attention.

## 3 METHODS

### 3.1 OVERVIEW

In the few-shot segmentation (FSS) task, we have a training set and a testing set with no labeling overlap. The testing set consists of a support set $\mathcal{S}$ and a query set $\mathcal{Q}$, which are from the same novel category (unseen in the training set). To set up a $N$-shot scenario, we randomly sample $N$ support images $\{\mathcal{I}_s^1, ..., \mathcal{I}_s^N\}$ with corresponding masks $\{\mathcal{Y}_s^1, ..., \mathcal{Y}_s^N\}$ from $\mathcal{S}$ to recognize the same-semantic region in the query image $\mathcal{I}_q \in \mathcal{Q}$.

During training, we follow the popular meta-learning scheme (Tian et al. (2020a); Pambala et al. (2021)) and construct a meta-task,*i.e.*, sampling $N$ support and one query from the training set into each episode. The proposed MuHS feature extractor is based on the transformer backbone consisting of $L$ transformer blocks, as illustrated in Fig. 2 (a). Given an input image, we split it into non-overlapping image patches, linearly embed them into patch tokens, and feed the patch tokens into the MuHS feature extractor. We utilize $X_q$, $X_s$ to denote the embedding of query tokens and support tokens, respectively. Specifically for the query image, we randomly discard some patch tokens. Based on the transformer backbone, MuHS has three major components as below:

1) MuHS enforces a Cross-Sample Attention between the support and query through some linking tokens. These linking tokens are trainable and update themselves by absorbing all query and support tokens simultaneously. In the following transformer block, the updated state of the linking tokens are absorbed by the query and support, respectively. Therefore, it facilitates interaction between the query and support with relatively low computational cost. The details are in Section 3.2.

2) Within each transformer block, MuHS appends an additional Cross-Region Interaction upon the original self-attention layer. While the original self-attention encourages interaction among similar

patch tokens, the Cross-Region Interaction promotes interaction among dissimilar regions in the background. The details are in Section 3.3.

3) Based on the output of the MuHS feature extractor, we use a matching head (*e.g.*, the linear classification head) to make partial prediction for the existing patches of the query image (note that some query patches are discarded at the input layer). Afterwards, we fill trainable mask tokens to the incomplete query patch tokens, and input them into a Masked Image Segmentation model consisting of $M$ transformer decoder. The Masked Image Segmentation aims to make holistic prediction of the input query, regardless of the discarded patches. The details are in Section 3.4.

## 3.2    CROSS-SAMPLE ATTENTION

The **Cross-Sample Attention** mechanism constructs interaction between support and query so as to suppress the sample-level heterogeneity, as illustrated in Fig. 2 (c). Specifically, we use linking tokens $X_{link} \in \mathcal{R}^{C \times D}$ ($C$ embedding vectors with $D$ dimensions ) to interact query and the support. At the input block, we initialize two linking tokens $X_{link}^0$ with the mean features of the foreground and background region of the support sample. Then the linking tokens are updated block by block through cross-attention, which is formulated as:

$$X_{link}^{i+1} = Att_{crs}(\texttt{Que}(X_{link}^i), \texttt{Key}(\{X_s^i, X_q^i, X_{link}^i\}), \texttt{Val}(\{X_s^i, X_q^i, X_{link}^i\})), \tag{1}$$

where $X_{link}^{i+1}$ denotes the updated linking tokens, $\{,\}$ denotes the concatenation operation. $\texttt{Que}$, $\texttt{Key}$, $\texttt{Val}$ are the operation to calculate $\texttt{query}$ embedding, $\texttt{key}$ embedding and $\texttt{value}$ embedding of support tokens $X_s^i$, query tokens $X_q^i$ and linking tokens $X_{link}^i$ in $i$-th MuHS Block, respectively.

Given the updated linking tokens, we then update the support and the query patch tokens by:

$$X_s^{i+1} = Att(\texttt{Que}(X_s^i), \texttt{Key}(\{X_s^i, X_{link}^{i+1}\}), \texttt{Val}(\{X_s^i, X_{link}^{i+1}\}))$$
$$X_q^{i+1} = Att(\texttt{Que}(X_q^i), \texttt{Key}(\{X_q^i, X_{link}^{i+1}\}), \texttt{Val}(\{X_q^i, X_{link}^{i+1}\})), \tag{2}$$

where $X_s^i, X_q^i$ are the $D$ dimensions embedding of support and query tokens in $i$-th Block. $X_{link}^{i+1}$ denotes the embedding of Linking tokens updated by Eq. 1.

Since the linking tokens already absorb information from all the query and support tokens (Eq. 1), in the subsequent Eq. 2, they propagate the absorbed information onto the support and the query tokens, therefore facilitating a mediate interaction between the support and query samples. Compared with directly constructing patch-to-patch attention between the query and support sample, our solution with linking tokens has the advantage of high efficiency. Specifically, the patch-to-patch attention incurs quadratic complexity against the patch token numbers, while using the linking tokens only incurs linear complexity.

## 3.3    CROSS-REGION INTERACTION

The **Cross-Region Interaction** (Fig. 2 (b)) is appended upon the self-attention layer in Eq. 2 within each MuHS block to encourage the interaction between different regions in the background. To this end, we use the ground-truth label to split the background in each query image (during training) into multiple regions. Specifically, some regions in the background actually belong to some annotated foreground classes but are merged into the background because the current training episode focuses on a different foreground class. We denote these regions as temporary background, and the remaining part (which has no foreground annotations) as constant background. Correspondingly, we use $\boldsymbol{x}^{[tb]}$ and $\boldsymbol{x}^{[cb]}$ to distinguish the tokens from the temporal and the constant background regions, respectively, and use $\boldsymbol{x}^{[f]}$ to represent the tokens in the foreground-of-interest in the current episode.

The Cross-Region Interaction compares the cosine distance between $\boldsymbol{x}^{[tb]}$ and $\boldsymbol{x}^{[cb]}$ after the self-attention layer (Eq. 2) and makes $\boldsymbol{x}^{[cb]}$ absorb information from the dissimilar $\boldsymbol{x}^{[tb]} \in X^{[tb]}$ by:

$$\boldsymbol{x}^{[cb]'} = \boldsymbol{x}^{[cb]} + softmax(1 - \frac{\boldsymbol{x}^{[cb]} \cdot X^{[tb]}}{|\boldsymbol{x}^{[cb]}| \cdot |X^{[tb]}|}) \cdot X^{[tb]}, \tag{3}$$

where $\boldsymbol{x}^{[cb]'}$ denotes updated constant background tokens embedding.

**Region-level triplet loss.** Besides the above Cross-Region Interaction smoothing the background across different regions through attention, we further use a region-level triplet loss to pull close the constant background tokens and temporal background tokens on the last MuHS transformer block. The region-level triplet loss is enforced on the final output state of the background tokens by:

$$\mathcal{L}_{tri} = max(\mathcal{D}(\boldsymbol{x}^{[tb]}, \boldsymbol{x}^{[cb]}) - \mathcal{D}(\boldsymbol{x}^{[tb]}, \boldsymbol{x}^{[f]}), 0)), \tag{4}$$

where $\mathcal{D}(.,.)$ is the cosine distance between two tokens.

## 3.4 MASKED IMAGE SEGMENTATION

The **Masked Image Segmentation** (MIS) model is appended upon the feature extractor of MuHS. It makes two types of prediction, *i.e.*, a partial prediction from a matching head and a holistic prediction from an additional "decoder + matching head". The details are as below:

• **Partial prediction**: We recall that MuHS randomly discards some patches of the input query image during training. Correspondingly, the output tokens $X_q^L$ from the $L$-layer MuHS feature extractor are incomplete. Given these existing output patches, we use a matching head (linear classification head) to make partial prediction in Fig. 2 (a). Specifically, we calculate the foreground / background mean features with the support tokens $X_s^L$ and the ground-truth object mask to correspondingly derive the foreground / background proxies $\boldsymbol{w}_f$ and $\boldsymbol{w}_b$. During training, the supervision on a specified query token $\boldsymbol{x}_q^L$ is:

$$\mathcal{L}_{partial}(\boldsymbol{x}_q^L) = -\log \frac{y_q \exp(\boldsymbol{w}_f^{\mathrm{T}}\boldsymbol{x}_q^L) + (1-y_q)\exp(\boldsymbol{w}_b^{\mathrm{T}}\boldsymbol{x}_q^L)}{\exp(\boldsymbol{w}_f^{\mathrm{T}}\boldsymbol{x}_q^L) + \exp(\boldsymbol{w}_b^{\mathrm{T}}\boldsymbol{x}_q^L)}, \tag{5}$$

where $y_q$ is corresponding label of the query token $\boldsymbol{x}_q^L$ ($y_q$=1, if $y_q$ belongs to the foreground; otherwise $y_q$=0).

• **Holistic prediction**: In addition, we fill mask tokens with trainable positional embeddings to construct the full set of the query patches (as shown in Fig. 2 (a)). Therefore, the output tokens $X_q^M$ from the Masked Image Segmentation model are complete. We make holistic prediction for all the query patches from an additional matching head, the weight matrix from which is calculated by foreground / background mean features with the support tokens $X_s^M$ from the Masked Image Segmentation model and the ground-truth object mask. The holistic prediction is supervised under cross-entropy loss $\mathcal{L}_{holistic}$, which is similar as in Eq. 5 and is thus omitted here.

We note that Masked Image Segmentation model is only utilized for training. During testing, we feed all the query patches into MuHS feature extractor and compare each query token against these proxies to classify each patch into the foreground / background.

**Overall Training.** During the training stage, we successively perform the feature extractor and Masked Image Segmentation model for the query prediction. The overall training loss is as follows:

$$\mathcal{L} = \mathcal{L}_{partial} + \alpha \cdot \mathcal{L}_{tri} + \beta \cdot \mathcal{L}_{holistic} \tag{6}$$

where $\alpha$ and $\beta$ are weighting factors.

## 4 EXPERIMENTS

### 4.1 DATASETS.

We evaluate the proposed MuHS on two datasets: PASCAL-$5^i$ (Shaban et al. (2017)) and COCO-$20^i$ (Nguyen & Todorovic (2019)). PASCAL-$5^i$ consists of PASCAL VOC 2012 (Everingham et al. (2010)) and additionally annotations from SDS (Hariharan et al. (2014)). We divide 20 classes into 4 splits and each split has 5 classes. During evaluation on one split (5 classes), we have other three splits (15 classes) for training. We randomly sample 1000 pairs of support and query in each split testing. COCO-$20^i$ is built from COCO2014 (Lin et al. (2014)). We divide 80 classes into 4 splits and each split has 20 classes. During evaluation on one split (20 classes), we have other three splits (60 classes) for training. We randomly sample 20000 pairs of support and query in each split testing.

Following previous works (Tian et al. (2020b); Zhang et al. (2021b)), we compare the performance on testing splits by using mean intersection over union (mIoU) as our evaluation metrics.

| Model | Method | 1-shot | | | | | 5-shot | | | | |
|-------|--------|------|------|------|------|------|------|------|------|------|------|
| | | S0 | S1 | S2 | S3 | Mean | S0 | S1 | S2 | S3 | Mean |
| Res50 | PGNet | 56.0 | 66.9 | 50.6 | 50.4 | 56.0 | 54.9 | 67.4 | 51.8 | 53.0 | 56.8 |
| | RPMM | 55.2 | 66.9 | 52.6 | 50.7 | 56.3 | 56.3 | 67.3 | 54.5 | 51.0 | 57.3 |
| | PFENet | 61.7 | 69.5 | 55.4 | 56.3 | 60.8 | 63.1 | 70.7 | 55.8 | 57.9 | 61.9 |
| | CyCTR | 67.8 | 72.8 | 58.0 | 58.0 | 64.2 | 71.1 | 73.2 | 60.5 | 57.5 | 65.6 |
| | HSNet | 64.3 | 70.7 | 60.3 | 60.5 | 64.0 | 70.3 | 73.2 | 67.4 | 67.1 | 69.5 |
| | BAM | 69.0 | 73.6 | 67.6 | 61.1 | 67.8 | 70.6 | 75.1 | 70.8 | 67.2 | 70.9 |
| Res101 | DAN | 54.7 | 68.6 | 57.8 | 51.6 | 58.2 | 57.9 | 69.0 | 60.1 | 54.9 | 60.5 |
| | RePRI | 59.6 | 68.6 | 62.2 | 47.2 | 59.4 | 66.2 | 71.4 | 67.0 | 57.7 | 65.6 |
| | PFENet | 60.5 | 69.4 | 54.4 | 55.9 | 60.1 | 62.8 | 70.4 | 54.9 | 57.6 | 61.4 |
| | CyCTR | 69.3 | 72.7 | 56.5 | 58.6 | 64.3 | 73.5 | 74.0 | 58.6 | 60.2 | 66.6 |
| | HSNet | 67.3 | 72.3 | 62.0 | 63.1 | 66.2 | 71.8 | 74.4 | 67.0 | 68.3 | 70.4 |
| DeiT-B | Baseline | 67.2 | 68.3 | 60.4 | 59.6 | 63.9 | 74.2 | 77.1 | 75.2 | 72.6 | 74.8 |
| | **Ours** | **71.2** | **71.5** | **67.0** | **66.6** | **69.1** | **75.7** | **77.8** | **78.6** | **74.7** | **76.7** |

Table 1: Comparison with the state of the arts on PASCAL-$5^i$ for 1-shot and 5-shot setting.

| Model | Method | 1-shot | | | | | 5-shot | | | | |
|-------|--------|------|------|------|------|------|------|------|------|------|------|
| | | S0 | S1 | S2 | S3 | Mean | S0 | S1 | S2 | S3 | Mean |
| Res50 | RePRI | 32.0 | 38.7 | 32.7 | 33.1 | 34.1 | 39.3 | 45.4 | 39.7 | 41.8 | 41.6 |
| | HSNet | 36.3 | 43.1 | 38.7 | 38.7 | 39.2 | 43.3 | 51.3 | 48.2 | 45.0 | 46.9 |
| | BAM | 43.4 | 50.6 | 47.5 | 43.4 | 46.2 | 49.3 | 54.2 | 51.6 | 49.6 | 51.2 |
| Res101 | DAN | - | - | - | - | 24.4 | - | - | - | - | 29.6 |
| | PFENet | 34.3 | 33.0 | 32.3 | 30.1 | 32.4 | 38.5 | 38.6 | 38.2 | 34.3 | 37.4 |
| | HSNet | 37.2 | 44.1 | 42.4 | 41.3 | 41.2 | 45.9 | 53.0 | 51.8 | 47.1 | 49.5 |
| DeiT-B | Baseline | 40.7 | 44.3 | 47.8 | 39.5 | 43.1 | 51.3 | 56.5 | 53.6 | 52.1 | 53.4 |
| | **Ours** | **44.0** | **50.0** | **49.1** | **46.3** | **47.4** | **53.6** | **60.5** | **57.3** | **55.2** | **56.7** |

Table 2: Comparison with the state of the arts on COCO-$20^i$ for 1-shot and 5-shot setting.

## 4.2 IMPLEMENTATION DETAILS

We adopt DeiT-B/16 (Touvron et al. (2021)) (pretrained on Imagenet (Deng et al. (2009))) as our backbone. We use SGD optimizer and set the learning rate as 9e-4. We randomly crop images to $480 \times 480$ and follow the data augmentation in PFENet (Tian et al. (2020b)). For PASCAL-$5^i$, we train 50 epochs with batch size 4. For COCO-$20^i$, we train 30 epochs and set batch size to 16. The proposed MuHS is trained on Pytorch with 4 NVIDIA A100 GPUS. More details are in Sec A.2.

## 4.3 COMPARISON WITH THE STATE OF THE ARTS

We compare MuHS with the existing state of the arts on PASCAL-$5^i$ and COCO-$20^i$. The results on two datasets are summarized in Table. 1 and Table. 2, respectively.

From Table. 1, we clearly observe the superiority of MuHS on PASCAL-$5^i$. First, comparing MuHS against the DeiT-B baseline, we find MuHS achieves considerable improvements. For example, under the 1-shot and 5-shot settings, MuHS outperforms the DeiT-B baseline by +5.2% and +1.9% mIoU, respectively. Second, MuHS surpasses all the state-of-the-art methods by a clear margin (especially under the 5-shot setting). For example, MuHS clearly outperforms the most competing method BAM (Lang et al. (2022)) by +1.3% and +5.8% mIoU on 1-shot and 5-shot, respectively. We note that all the competing methods use sophisticated matching heads, while our MuHS only uses a simple linear classification head. We thus infer that the superiority of our MuHS mainly comes from the discriminative features, indicating that MuHs is a strong feature extractor for FSS.

The observation on COCO-$20^i$ (Table. 2) is similar, *i.e.*, MuHS improves the DeiT-B baseline and presents superiority against all the competing methods. For example, under the 5-shot settings, MuHS surpasses the DeiT-B baseline by +3.3% mIoU and surpasses BAM (Lang et al. (2022)) by +5.5%, mIoU, respectively.

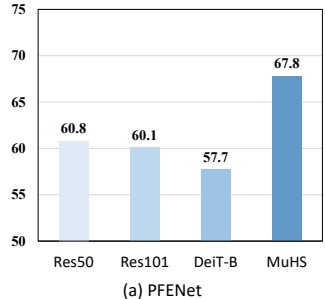 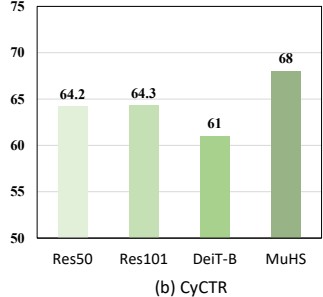

Figure 3: MuHS feature is compatible to multiple FSS matching heads

| Components | PASCAL-$5^i$ | | COCO-$20^i$ | |
|---|---|---|---|---|
| | 1-shot | 5-shot | 1-shot | 5-shot |
| Baseline | 63.9 | 74.8 | 43.1 | 53.4 |
| Baseline + Cross-Sample Attention | 65.8 | 75.3 | 45.8 | 54.7 |
| Baseline + Cross-Region Interaction | 66.3 | 75.5 | 46.1 | 54.8 |
| Baseline + Masked Image Segmentation | 66.9 | 75.8 | 46.3 | 55.1 |
| **All (MuHS)** | **69.1** | **76.7** | **47.4** | **56.7** |

Table 3: Ablation studies of our proposed method under 1-shot and 5-shot setting.

## 4.4 COMPATIBILITY TO STATE-OF-THE-ART MATCHING HEADS.

Since the proposed MuHS focuses on learning the feature extractor, we are interested at its compatibility to different FSS matching heads. Specifically, we investigate two popular FSS heads, *i.e.*, PFENet (Tian et al. (2020b)) and CyCTR (Zhang et al. (2021b)). We compare our MuHS feature against two frozen CNN feature (*i.e.,* ResNet50, ResNet101, following their original practice), as well as the DeiT-B feature. The results are summarized in Fig. 3, from which we draw three observations as below:

First, compared with the frozen CNN features, the frozen DeiT-B features considerably decreases accuracy for both the PFENet and CyCTR heads. For example, the frozen DeiT-B feature is inferior than the frozen ResNet50 feature by $\downarrow 3.1\%$ , $\downarrow 3.2\%$ mIoU on PFENet and CyCTR, respectively. It is somehow surprising given that the transformer feature usually achieves better discriminative ability than the CNN counterparts. However, we think the above observation is actually reasonable, because both the PFENet and CyCTR heads (as well as most matching heads in prior literature) are specifically designed for CNN features and lack consideration for the transformer features.

Second, our MuHS features significantly outperform the CNN features, although the employed heads is specifically designed for the CNN features. For example, using the PFENet head, MuHS feature surpasses the ResNet101 feature by $\uparrow 7.70\%$. We infer that although there are some weaknesses for cooperating the transformer feature with these heads (as in observation 1), the benefits of suppressing the heterogeneity in MuHS dominate. Therefore, MuHS brings improvement over the CNN features and shows good compatibility against these matching heads.

Third, comparing the achieved mIoU of "MuHS + PFENet (CyCTR) head" against the "MuHS + linear classification head" (in Table. 1), we find that the latter is slightly better. This result is consistent with the above two observations: MuHS is the superior feature for the PFENet and CyCTR heads (observation 2), while those two heads are not the superior heads for MuHS and other transformer features (observation 1). Therefore, we recommend using the simple linear classification head for MuHS currently. That being said, we are optimistic towards a good embrace between MuHS and future matching heads in the FSS community.

## 4.5 ABLATION STUDIES

MuHS adopts Cross-Sample Attention, Cross-Region Interaction and Masked Image Segmentation to reinforce the interaction between different samples, regions and neighboring patches and suppress

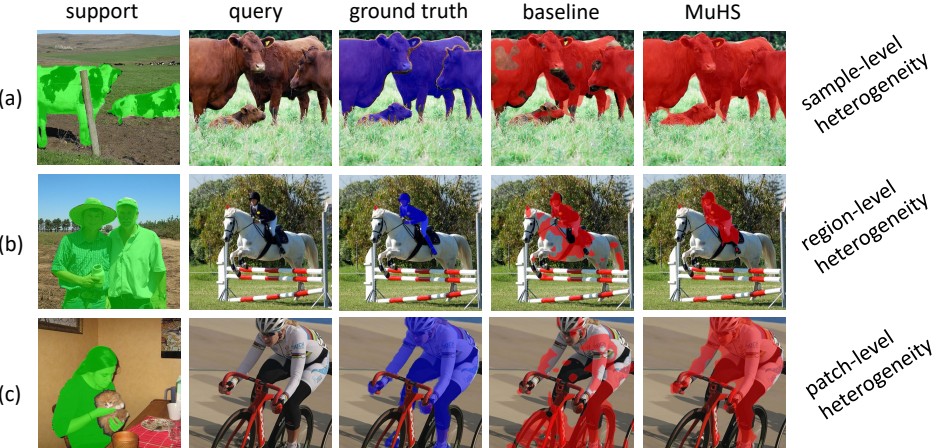

Figure 4: Visualization of the segmentation results on PASCAL-$5^i$. (a) MuHS suppresses the sample-level heterogeneity between cattle (query) and cow (support), therefore improving the recall of the "cattle" pixels in the query. (b) MuHS suppresses the region-level heterogeneity and merges "horse" into the background of the query, when the support object is "person". (c) Due to its capacity of suppressing the patch-level heterogeneity, MuHS recognizes the upper part of body, although there is no such a clue (*i.e.*, white upper body) from the support.

the heterogeneity in FSS. We investigate their benefits through ablation, as shown in Table 3. We draw two observations:

First, all the three modules can bring considerable improvements, independently. For example, under 1-shot on PASCAL-$5^i$, adding Cross-Sample Attention, Cross-Region Interaction and Masked Image Segmentation alone improves the baseline by +1.9%, +2.4% and +3.0%, respectively.

Second, MuHS integrating all the three components achieves further improvement, *e.g.,* +5.2% on PASCAL-$5^i$ under 1-shot setting. It indicates that these three components suppressing different heterogeneity achieve complementary benefits. We supplement more ablation studies in Sec A.5.

**Visualization of heterogeneity suppression.** We visualize some segmentation results in Fig. 4 to intuitively understand how the proposed MuHS suppresses the heterogeneity and improves FSS. In Fig. 4 (a), due to the sample-level heterogeneity (between cattle and cow), the baseline fails to recall many foreground pixels. In contrast, the proposed MuHS significantly improves the recall on the foreground pixels due to its capacity of suppressing the sample-level heterogeneity. In Fig. 4 (b), the region-level heterogeneity makes the baseline to classify many pixels on the horse into the foreground (person). In contrast, MuHS smooths the background and thus remove the distraction from the horse. In Fig. 4 (c), MuHS successfully merges the upper part of body into the foreground "person" by suppressing patch-level heterogeneity. Additionally, we visualize the feature distribution before and after eliminating heterogeneity in Sec A.4 and plot the convergence curves in Sec A.6.

## 5 CONCLUSION

This paper proposes a feature extractor with Multi-level Heterogeneity Suppressing (MuHS) for few-shot semantic segmentation(FSS). Based on the transformer backbone, MuHS sets up novel cross-sample attention, cross-region interaction and the masked image segmentation. The cross-sample attention efficiently propagates information across different samples. The cross-region interaction facilitates feature absorption between dissimilar regions within the background. The masked image segmentation utilizes the contextual information to infer the labels for discarded (masked) patch tokens so as to reinforce the capacity of contextual inference. These three modules respectively suppress the heterogeneity from three different levels, therefore improving the intra-class compactness of the FSS features. Extensive experiments on two popular FSS datasets demonstrate the effectiveness of MuHS and the achieved results are on par with the state of the art.

## ETHICS STATEMENT

This paper can help to improve the semantic segmentation accuracy with the limited labeled samples. It may be applied to automatic driving system to improve safety when the system needs to recognize unseen objects. We will explore more applications of few-shot segmentation. Moreover, we will try to improve the reliability of few-shot segmentation systems to reduce potential problems.

## REPRODUCIBILITY STATEMENT

The MuHS is reproducible. In the main text, we describe the two datasets for evaluation, (*i.e.,* PASCAL-$5^i$ and COCO-$20^i$) and the details about the experimental implementation. We supply the analysis of some hyper parameters in appendix.

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

# A   APPENDIX

In the appendix, we supply the details which are not described in the main text due to space limitation. In Section A.1, we analyze the impact of some hyper-parameters. In Section A.2, we add more implementation details. In Section A.3, we adopt one more dataset to evaluate performance. In Section A.4, we compare the feature distribution before and after eliminating heterogeneity. In Section A.5, we supply more ablation experiments. In Section A.6, we plot the convergence curves of the proposed MuHS and recent state-of-the-art methods.

## A.1   Hyper-parameters analysis.

We analyze the impact of important hyper-parameters, *i.e.*, $\alpha$ and $\beta$ in Eq. 6. and investigate the impact of model depth and masked ratio of the proposed Masked Image Segmentation (MIS) in Section 3.4. The experiments are reported on split-0 of PASCAL-$5^i$, under 1-shot setting.

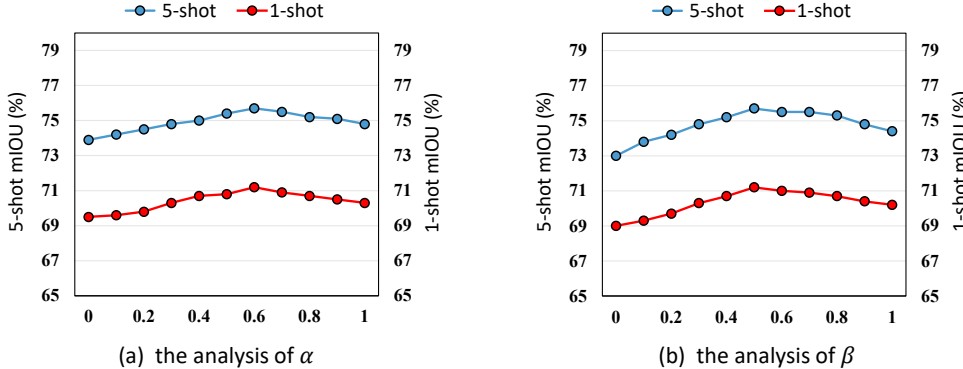

(a)  the analysis of $\alpha$         (b)  the analysis of $\beta$

Figure 5: Analysis on the hyper-parameters.

In Fig. 5 (a), we evaluate the impact of $\alpha$, which controls the weight of triplet loss in Eq. 6. We observe that the accuracy first increases (when $\alpha$ increases from 0 to 0.6) and then decreases (when $\alpha$ further increases to 1.0). We set $\alpha = 0.6$ as the weight factor.

In Fig. 5 (b), we evaluate the impact of $\beta$, which controls the weight of holistic prediction loss in Eq. 6. We observe that the accuracy first increases (when $\beta$ increases from 0 to 0.5) and then decreases (when $\beta$ further increases to 1.0). We set $\beta = 0.5$ as the weight factor.

| MIS layers | mIoU |
|:---:|:---:|
| 5 | 70.5 |
| 6 | 70.7 |
| **7** | **71.2** |
| 8 | 70.8 |

| Mask ratio | mIoU |
|:---:|:---:|
| 3% | 70.4 |
| 5% | 70.9 |
| **7%** | **71.2** |
| 9% | 71.0 |

Table 4:  Analysis on the MIS model depth

Table 5:  Analysis on the masked ratio for MIS

In Table 4, we analyse the impact of Masked Image Segmentation (MIS) model depth. We observe that the 7-layer MIS model can achieve the best accuracy. In Table 5, we analyse the impact of masked ratio for MIS model. We observe that randomly masking out 7% patches can achieve the best accuracy.

## A.2   Implementation details.

We supply the implementation details of how to transform a patch token into pixel-wise mask map and the scheme to generate the discarded patches.

• To transform a patch into pixel-wise mask map, we follow the common practice (PANet (Wang et al. (2019)), PFENet (Tian et al. (2020b))) of spatially up-sampling. Specially, we first obtain the softmax scores for each patch token through the classification head. Then, the score maps are up-sampled through bi-linear interpolation to match the size of the input image. Finally, we use argmax operation to generate pixel-wise mask map.

• To generate the discarded patches, we randomly shuffle the patch tokens and then mask the rear of the token sequence. This operation is the same as in other MIM methods (*e.g.*, MAE (He et al. (2022)) ).

### A.3 EVALUATION ON MORE DATASETS.

We evaluate one more dataset, i.e., Cityscapes (Cordts et al. (2016)), an urban street-scene dataset. We use 15 classes (out of the commonly-used 19 classes) to construct the base set and use the other 4 classes (i.e., "sky", "person", "car", "bicycle") for the novel set.

|  | Baseline | CyCTR | MuHS |
|---|---|---|---|
| Cityscapes | 13.1 | 15.2 | 25.2 |

Table 6: Comparison with the state of the art and the baseline on cityscapes for 1-shot setting.

Based on this newly-generated few-shot segmentation benchmark, we compare the proposed MuHS against the baseline and a most recent state-of-the-art method CyCTR(Zhang et al. (2021b)). From the Table 6, we observe MuHS significantly surpasses the baseline and CyCTR by +12.1% and +10% mAP, respectively. We note that the cityscapes is challenging for few-shot segmentation task. We infer it is because in cityscapes, the number of semantic classes appearing in a single image is much larger, therefore increasing the challenge from region-level heterogeneity.

### A.4 COMPARISON OF THE FEATURE DISTRIBUTION.

To better understand how MuHS suppresses the three types of heterogeneity, we use t-SNE visualization to compare the feature distribution before and after MuHS in Fig. 6.

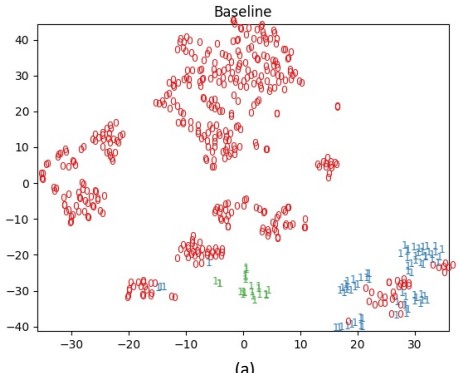 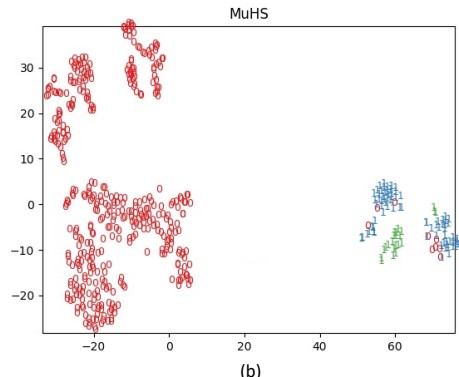

Figure 6: Visualization of the feature distribution of the baseline and MuHS. We evaluate the class of "person". Red and Blue denotes the background and foreground features in the query, respectively. We additionally use Green to denote foreground features of the support to visualize the distribution gap between the support and the query.

We correspondingly draw two observations. First, the intra-class distributions of both background and the foreground become more compact, indicating that MuHS effectively suppresses the region-level and patch-level heterogeneity. Second, the foreground from the support and query images become closer, indicating that MuHS effectively suppresses the sample-level heterogeneity. These observations are consistent with the segmentation examples in Fig. 4. This experiment also intuitively validates that MuHS improves intra-class feature compactness.

A.5 ABLATIONS ON MORE VARIANTS.

We recall that all the three components (*i.e.*, cross-sample attention, cross-region interaction and masked image segmentation) can bring considerable improvements, independently and integrating all of them achieves further improvements. For better investigating the efficiency, we supply more ablation studies by adding three more variants, as shown in Table 7. Each variant combines two components out of cross-sample attention, cross-region interaction and masked image segmentation can still improve the baseline.

| Cross-Sample Atten | Cross-Region Inter | MIS | PASCAL-$5^i$ |
|:---:|:---:|:---:|:---:|
| | | | 63.9 |
| ✓ | | | 65.8 |
| | ✓ | | 66.3 |
| | | ✓ | 66.9 |
| ✓ | ✓ | | 67.4 |
| ✓ | | ✓ | 67.8 |
| | ✓ | ✓ | 68.2 |
| ✓ | ✓ | ✓ | 69.1 |

Table 7: Ablation studies on more variants of three components for PASCAL-$5^i$.

A.6 CONVERGENCE CURVES.

In Fig. 7, we plot the convergence curves of CyCTR, PFENet and MuHS. Compared with CyCTR and PFENet, MuHS costs 4 × less training epochs (50 epochs) with fewer time (8 hours) and achieves higher accuracy on PASCAL-$5^i$. The comparison is based on Pytorch and NVIDIA A100 GPU.

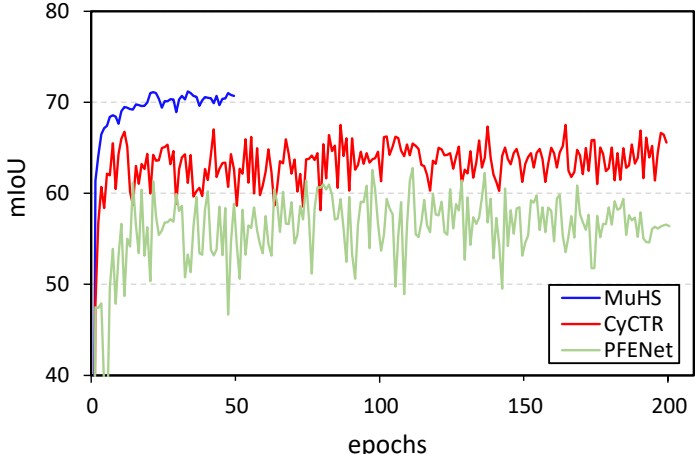

Figure 7: The convergence curves of recent methods and MuHS on split-0 of PASCAL-$5^i$.

