# OpenReview forum: "Suppressing the Heterogeneity: A Strong Feature Extractor for Few-shot Segmentation"
_ICLR.cc/2023/Conference — ICLR 2023 poster_

### Official Review · Reviewer_FHxi · 2022-10-21

**Confidence:** 4
**Correctness:** 3
**Technical Novelty And Significance:** 3
**Empirical Novelty And Significance:** 3
**Recommendation:** 6

**Clarity, Quality, Novelty And Reproducibility:**

The quality of the work is very good, and it also provides new ideas for the subsequent work. The introduction of the overall structure and each module is also clear and original.

**Strength And Weaknesses:**

Strength ：
The feature extractor has been overlooked by recent state-of-the-art methods. Accurately analyze the heterogeneity from three levels in the FSS task. This paper propose a feature extractor with Multi-level Heterogeneity Suppressing (MuHS).
The paper according to the heterogeneity has three levels from coarse to fine leverages the attention mechanism in transformer backbone to effectively suppress all these three-level heterogeneity. Experimental results confirm that MuHS is compatible to multiple FSS heads and achieves new state of the art using a simple linear classification head.

Weaknesses：
Instead of designing a special classification header based on such a powerful feature extractor, a simple linear classification header is used.


**Summary Of The Paper:**

This paper tackles the Few-shot Semantic Segmentation (FSS) task with focus on learning the feature extractor. This paper thinks the FSS feature extractor deserves exploration and observe the heterogeneity (i.e., the intra-class diversity in the raw images) as a critical challenge hindering the intra-class feature compactness. In this paper, aothors propose Multi-level Heterogeneity Suppressing (MuHS), which utilizes novel cross sample attention, cross-region interaction and masked image segmentation to suppress the heterogeneity from three levels.


**Summary Of The Review:**

1、Can you provide a visual comparison of the feature maps before and after eliminating heterogeneity？Please add relevant experiments to make the paper more comprehensive.
2、Would the results be better if the classification headers were replaced with those of dense corresponding methods( (i.e., HSNet, DCAMA))?
3、The comparison results of convergence speed can be added.
4、The number of learnable parameters required by different methods also needs to be compared.
5、One of the goals of the article is to achieve the intra-class feature compactness. But subsequent experiments don’t prove this.

---

> ### Author Response · Authors · 2022-11-18
> **Response to Reviewer FHxi**
>
> **FHxi-Q1: Can you provide a visual comparison of the feature maps before and after eliminating heterogeneity? Please add relevant experiments to make the paper more comprehensive.**
>
> **[Ans]:** Thanks for this good suggestion. Visualizing the feature distribution can help us to better understand how MuHS suppresses the three types of heterogeneity. Therefore, we follow your suggestion and use t-SNE visualization to compare the feature distribution before and after MuHS in the appendix (A.4.).
>
> We correspondingly draw two observations. First, the intra-class distributions of both background and the foreground become more compact, indicating that MuHS effectively suppresses the region-level and patch-level heterogeneity. Second, the foreground from the support and query images become closer, indicating that MuHS effectively suppresses the sample-level heterogeneity. These observations are consistent with the segmentation examples in Fig. 4 in the manuscript.
>
> This experiment also intuitively shows that MuHS improves intra-class feature compactness (as mentioned in your 5th question). We appreciate this insightful suggestion and have correspondingly revised the manuscript.
>
>
> * * *
>
> **FHxi-Q2:  A weakness is this paper uses only a simple linear classification header, instead of designing a special classification head. Would the results be better if the classification headers were replaced with those of dense corresponding methods (i.e., HSNet, DCAMA)?**
>
> **[Ans]:**  We respectfully hope it will not be considered as a weakness that we did not design a special classification head. As we state in the abstract and introduction, this paper intends to shift the focus from designing the matching head to designing a FSS feature extractor. While prior state-of-the-art methods have intensively explored the FSS head, we draw an attention that the feature extractor is also important.
>
> Though we did not design new FSS head, we have employed multiple already-existing FSS heads to cooperate with our feature extractor (MuHS). Experimental results (Fig. 3 in the manuscript) show that our feature extractor brings consistent improvement to popular FSS heads. For example, when the FSS head is from PFENet (CyCTR), the MuHS feature outperforms the popular ImageNet-pretrained ResNet101 feature by +7.7% (+3.7%) mAP. Moreover, in the manuscript, we find linear classification head is the best choice for our MuHS feature extractor, though all the heads gain improvement from MuHS.
>
> That being said, we take your suggestion on "dense corresponding" methods seriously. We find that the dense corresponding in two recommended methods (HSNet and DCAMA) is used to generate prior mask and is not directly applicable to our MuHS (because MuHS does not need prior map). We guess your actual interest might be using dense corresponding strategy for the classification head. Therefore, we duplicate the pixel-to-pixel comparison in the mentioned methods and apply it for pixel-wise classification. It achieves comparable results (68.6% vs. 69.1%) as the linear classification head. We infer it is because MuHS already well suppresses the heterogeneity and achieves good intra-class feature compactness, so that a linear classification is already sufficient.
>
>
> * * *
>
> **FHxi-Q3: The comparison results of convergence speed can be added.**
>
> **[Ans]:** Thanks. During rebuttal, we compare MuHS against  recent state-of-the-art method CyCTR and PFENet regarding the convergence speed. The convergence curves are plotted in the appendix (A.6). It is observed that MuHS costs 4 × less training epochs (50 epochs) with fewer time (8 hours) and achieves higher accuracy on PASCAL-5$^{i}$. The above comparison is based on Pytorch and NVIDIA A100 GPU.
>
> * * *
>
> **FHxi-Q4: The number of learnable parameters required by different methods also needs to be compared.**
>
> **[Ans]:**  Three recent state-of-the-art methods, i.e., PFENet, CyCTR and BAM, respectively have 53M, 59M and 71M learnable parameters. They only learn the matching head and freeze the ImageNet-pretrained feature extractor. In contrast, MuHS learns the feature extractor and has 137M learnable parameters.
>
>
> * * *
>
> **FHxi-Q5: One of the goals of the article is to achieve the intra-class feature compactness. But subsequent experiments don't  prove this.**
>
> **[Ans]:** Fig. 4 in the manuscript already have provided some intuitive examples showing that MuHS suppresses heterogeneity and thus improves the intra-class feature compactness. During the rebuttal, according to your suggestion in the first question, we use t-SNE visualization to compare the feature distribution before and after MuHS. It further confirms that MuHS enhances the intra-class feature compactness.

---

### Official Review · Reviewer_UVUC · 2022-10-23

**Confidence:** 4
**Correctness:** 4
**Technical Novelty And Significance:** 3
**Empirical Novelty And Significance:** 3
**Recommendation:** 6

**Clarity, Quality, Novelty And Reproducibility:**

The technical novelty is limited in the intuition that it is just a superposition of mature methods. The cross-attention, triplet loss, and masked image segmentation are mature methods.

**Strength And Weaknesses:**

Strength:
1. This is a good engineering job. Few-shot semantic segmentation is an important but challenging problem. The proposed components in the framework could be useful utilities in various applications.
2. The paper is well structured with clear intuition and explanation.

Weakness
1. Lack of novelty. The improvement is a bit incremental, although I do acknowledge the contribution of three components to few-shot segmentation. The main contribution is the application of some tricks in the few-shot segmentation task, which is an empirical advancement and the theoretical novelty is limited.
2. Although the paper has provided extensive evaluations over different metrics and datasets, some details of the experiment setup and results are not fairly convincing. For example, the setting of ablation studies is not appropriate, because an ablation study typically refers to removing some “feature” of the model or algorithm, and seeing how that affects performance. The results of ablation studies are not enough to validate the efficiency of the proposed components.

**Summary Of The Paper:**

In this work, the authors propose a new feature extraction framework called MuHS to improve few-shot semantic segmentation using multi-coarse heterogeneity-aware optimization. The MuHS is optimized with three key components: cross-sample attention, cross-region interaction, and masked image segmentation. The empirical results are sound to validate the efficiency of the proposed methods.

**Summary Of The Review:**

This is a good engineering job. But the novelty is limited.

---

> ### Author Response · Authors · 2022-11-18
> **Response to Reviewer UVUC**
>
> **UVUC-Q1: Lack of novelty. The improvement is a bit incremental, although I do acknowledge the contribution of three components to few-shot segmentation. The main contribution is the application of some tricks in the few-shot segmentation task, which is an empirical advancement and the theoretical novelty is limited.**
>
> **[Ans]:** We respectfully disagree with this point. These components (that you mentioned) only make up one of our three contributions. We would like to highlight our three major contribution as below:
>
> First, while most state-of-the-art FSS literatures focus on the matching head, this work shifts the focus to the feature extractor. We break a popular assumption that the FSS data is insufficient for learning a FSS feature extractor. Instead, we show that the true obstacle is that the few-shot setting amplifies the heterogeneity problem.
>
> Second, based on this insight, our method (MuHS) consisting of these three components (i.e., the cross-sample attention, cross-region interaction and masked image segmentation) is our second contribution.
>
> Third, we show that MuHS can serve as a general feature extractor for various FSS heads. Specifically, MuHS brings consistent improvement to multiple popular FSS heads, e.g., MuHS outperforms the popular ImageNet-pretrained ResNet101 feature extractor by  +7.7% (+3.7%) mAP when cooperating with PFENet (CyCTR) head (Fig. 3 in the manuscript). We note that this general improvement has NO conflict against the result that MuHS favors the linear classification head.
>
>
> * * *
>
> **UVUC-Q2: Although the paper has provided extensive evaluations over different metrics and datasets, some details of the experiment setup and results are not fairly convincing, e.g., the ablation study is not comprehensive enough.**
>
> **[Ans]:** Thanks for your reminding. We guess your concern on the ablation study is that we did not present the results of combining 2 components. Therefore, during rebuttal, we supplement the original ablation study with three variants. Each variant combines two components out of cross-sample attention, cross-region interaction and masked image segmentation in the appendix (A.5).

---

### Official Review · Reviewer_ZBET · 2022-10-24

**Confidence:** 2
**Correctness:** 3
**Technical Novelty And Significance:** 2
**Empirical Novelty And Significance:** 2
**Recommendation:** 5

**Clarity, Quality, Novelty And Reproducibility:**

The quality of the paper is good. It is a well written paper. A problem of interest to the community is tackled in the paper. The main idea of the paper is novel and is intuitivley sound.


Reproducibility: Codes have not been released.

**Strength And Weaknesses:**

The paper has the following main strengths:


1) The paper is well written and the main ideas are effectively presented in the paper.
2) The paper tackles a problem of interest to the community in FFS.
3) The paper presents a strong idea and is intuitively sound.
4) The paper shows significant improvemnet on the previous stat of the art.

The major weaknesses of the papers in my opinion are as follows:
1)The mehtod is only tested on two datasets. Have the authors tried more datasets to get a better idea of the performance.
2) The codes for the paper are not released.

**Summary Of The Paper:**

The authors propose training feature extractor for Few show segmentation. To suppress the sample level, region levle and patch level heterogeneity between samples, the author propose Multi-level Heterogeneity Suppressing extractor (MuHS). MuHS uses attention  to suppress the heterogeneities by reinforcing the attention between query and support samples, regions and patches, and a novel masked image segmentation.




**Summary Of The Review:**

The paper proposes to use transformers to learn feature extractor for FFS. The authors claim that feature extractors have been overlooked in the FFS literature and off the shelf networks trained on imagenet are used as feature extractors. The authors propose to use transformers to supress inter-sampels, region and patch heterogeniety in FFS.

The method is tested on two datasets and the resutlts show significant improvement.

I would like the authors to address the following:
1)The mehtod is only tested on two datasets. Have you tried more datasets to get a better idea of the performance.
2) The codes for the paper are not released.

---

> ### Author Response · Authors · 2022-11-18
> **Response to Reviewer ZBET**
>
> **ZBET-Q1: The method is only tested on two datasets. Have the authors tried more datasets to get a better idea of the performance.**
>
> **[Ans]:** Thank you for this suggestion. We note that our choice of datasets is standard in few-shot segmentation  (all the prior methods under this topic use these two datasets for evaluation).
>
> That being said, during rebuttal, we add one more dataset, i.e., Cityscapes$^{[4]}$, an urban street-scene dataset. We use 15 classes (out of the commonly-used 19 classes) to construct the base set and use the other 4 classes (i.e., sky, person, car, bicycle) for the novel set. Based on this newly-generated few-shot segmentation benchmark, we compare the proposed MuHS against the baseline and a most recent state-of-the-art method CyCTR$^{[5]}$ in the Table below.
>
> |  |Baseline  | CyCTR | MuHS |
> | --- | --- | --- | --- |
> | Cityscapes | 13.1 | 15.2 | 25.2 |
>
> It is observed that MuHS achieves 25.2% mAP and surpasses the baseline and CyCTR by +12.1% and +10% mAP, respectively. We also note that all the results are relatively low, compared with the results on other two datasets. We infer it is because in Cityscapes, the number of semantic classes appearing in a single image is much larger, therefore increasing the challenge from region-level heterogeneity. We add the performance of cityscapes in the appendix (A.3).
>
>
>
> [4] Marius Cordts, Mohamed Omran, Sebastian Ramos, Timo Rehfeld, Markus Enzweiler, RodrigoBenenson, Uwe Franke, Stefan Roth, and Bernt Schiele. The cityscapes dataset for semanticurban scene understanding. In Proc. of the IEEE Conference on Computer Vision and PatternRecognition, 2016.
>
> [5] Gengwei Zhang, Guoliang Kang, Yi Yang, and Yunchao Wei. Few-shot segmentation via cycle-consistent transformer. Advances in Neural Information Processing Systems, 34:21984–21996, 2021.
>
>
> * * *
>
> **ZBET-Q2: The codes for the paper are not released.**
>
> **[Ans]:** We provide the code of the three major components of MuHS in the supplementary and will release the official implementation upon acceptance.

---

> ### Author Response · Authors · 2022-12-07
> **Followup response to reviewer ZBET**
>
> Dear reviewer ZBET，
>
> We would like to thank you again for your suggestions. In our responses, we have made every effort to address your concerns. Specifically, 1) we add few-shot experiments on Cityscapes, in addition to the commonly-adopted FSS datasets. Experimental results further confirm our improvement over the baseline and the superiority against some competing methods. 2) We have uploaded the code of the key components into the supplementary. We promise that we will release the entire code upon acceptance. We sincerely hope you will find our paper acceptable based on our responses and welcome your further suggestions.
>
> Best.
>
> Authors

---

### Official Review · Reviewer_B9Ww · 2022-11-03

**Confidence:** 4
**Correctness:** 4
**Technical Novelty And Significance:** 3
**Empirical Novelty And Significance:** 4
**Recommendation:** 8

**Clarity, Quality, Novelty And Reproducibility:**

The idea of suppressing the intra-class variation to cope with heterogeneity is quite interesting for FSS problem. The proposed 3 modules (i.e. cross-sample, cross-region, MIS) seems to be reasonable. The organization of this paper is clear. I believe it is not difficulty to reproduce the proposed method.


**Strength And Weaknesses:**

Strength:
(1) Investigation of the FSS problem from the perspective of feature extraction, which is interesting and is ignored in previous research.
(2) Summarizing the heterogeneity between support-query pair and within query image itself from 3 aspects: sample-level, region-level and patch level is a good idea.
(3) The idea of leverage cross-sample attention, cross-region interaction and masked image segmentation to suppress the intra-class variation is reasonable and rather novel.
(4) The proposed method advances the state-of-the-art results.

Weaknesses:
(1) The number of “linking token” is set to 2 in this paper (i.e. foreground and background) manually. However, the authors claim there exists heterogeneity in different regions of background. Is it reasonable to only set 1 token for background?
(2) The demonstration of some notions are obscure, e.g. the dimension of   X_s^i and X_q^i should be explicitly pointed out for better understanding.
(3) some details are missing or not explained.  e.g., how to transform a patch into pixel-wise mask map? What is the principle to generate discarded patches?




**Summary Of The Paper:**

Different from most existing methods, this paper investigates the FSS problem from the perspective of feature extraction. The authors claim that the heterogeneity of the sample-level, region-level and patch-level of support-query pair is the main obstacle to alleviate the intra-class variation. Thus, this paper proposes 3 novel modules (i.e. cross-sample attention, cross-region interaction and masked image segmentation) to tackle the above 3 problems, respectively. The proposed method sets new states of the art on many benchmarks.



**Summary Of The Review:**

The proposed method is interesting and rather novel, achieving appealing results. The paper is also well written and easy to follow.

---

> ### Author Response · Authors · 2022-11-18
> **Response to Reviewer B9Ww**
>
> **B9Ww-Q1: Is it reasonable to only set 1 linking token for background? Since there exists heterogeneity in different regions of background, using more linking tokens might be better.**
>
>  **[Ans]:** Insightful question. During rebuttal, we tried 2, 4, 6 background linking tokens for cross-sample attention and find they indeed bring slight improvement. While using 1 linking token already improves the baseline by +2.4% mAP on PASCAL-$5^i$ under the 1-shot setting (Table 3 in the manuscript), adding more tokens further brings around +0.2% mAP improvement. Given that our original implementation (1 token) is relatively concise, we think using only 1 background linking token is acceptable.
>
> Specifically, we partition the input image to 1 $\times$ 2 / 2  $\times$ 2 / 2  $\times$ 3 (vertical/horizontal) parts and correspondingly gets 2 / 4 / 6 background regions. We use the mean feature of each part to initialize a respective background linking token. We find these settings achieve very close results (i.e., about 69.2 to 69.3 % mAP on 1-shot PASCAL-$5^i$), and are higher than our original implementation (69.1%) by around 0.2%. In other words, using only 1 background linking token already exploits the major benefit from cross-sample linking (interaction). We conjecture it is because the cross-region interaction in MuHS effectively suppresses the heterogeneity among different regions, therefore making 1 linking token representative for the background. That being said, we note that our current method for partitioning the background is relatively simple. We guess a more sophisticated partition strategy might be better. Therefore, we consider this suggestion valuable and will elaborately explore it in our future work.
>
> * * *
>
>  **B9Ww-Q2:  Some notions are obscure, e.g., the dimension of $X_s^i$ and $X_q^i$ should be explicitly pointed out for better understanding.**
>
> **[Ans]:** Thanks. We apologize for making some obscure notions. We have explicitly pointed out the dimension in the revised manuscript and will go through all the notions carefully.
>
> * * *
>
> **B9Ww-Q3: Some details are missing or not explained. e.g., how to transform a patch into pixel-wise mask map? What is the principle to generate discarded patches?**
>
> **[Ans]:**  Thanks. We add the mentioned implementation details in the appendix (A.2). These two details are as below:
>
> 1) To transform a patch into pixel-wise mask map, we follow the common practice (PANet$^{[1]}$, PFENet$^{[2]}$) of spatially up-sampling. Specially, we first obtain the softmax scores for each patch token through the classification head. Then, the score maps are up-sampled through bi-linear interpolation to match the size of the input image. Finally, we use argmax operation to generate pixel-wise mask map.
>
> 2) To generate the discarded patches, we randomly shuffle the patch tokens and then mask the rear of the token sequence. This operation is the same as in other MIM methods (e.g., MAE$^{[3]}$).
>
>     [1] Kaixin Wang, Jun Hao Liew, Yingtian Zou, Daquan Zhou, and Jiashi Feng. Panet: Few-shot image semantic segmentation with prototype alignment. In Proceedings of the IEEE/CVF International Conference on Computer Vision, pp. 9197–9206, 2019
>
>     [2] Zhuotao Tian, Hengshuang Zhao, Michelle Shu, Zhicheng Yang, Ruiyu Li, and Jiaya Jia. Prior guided feature enrichment network for few-shot segmentation. IEEE transactions on pattern analysis and machine intelligence, 2020.
>
>     [3] Kaiming He, Xinlei Chen, Saining Xie, Yanghao Li, Piotr Doll ́ar, and Ross Girshick. Masked au-toencoders are scalable vision learners. In Proceedings of the IEEE/CVF Conference on ComputerVision and Pattern Recognition, pp. 16000–16009, 2022.

---

### Author Response · Authors · 2022-11-18
**General Response**

We thank all the reviewers for their valuable comments. According to the suggestions, we have updated our manuscript with some more experiments, discussions and necessary revisions. We highlight these updates with blue color in the manuscript and summarize the main updates as below:

1. We use one more dataset, i.e., Cityscapes, to further evaluate the proposed MuHS, in the appendix (A.3).

2. We improve the ablation study by adding three more variants. Each variant combines two components out of Cross-Sample Attention, Cross-Region Interaction and Masked Image Segmentation in the appendix (A.5).

3. We compare the feature distribution before and after eliminating heterogeneity (i.e., baseline and the proposed MuHS) through t-SNE visualization in the appendix (A.4).

4. We plot the convergence curve of the proposed MuHS and recent state-of-the-art methods in the appendix (A.6).

5. We add more implementation details (e.g., how to transform a patch into pixel-wise mask map, how to generate the discarded patches) in the appendix (A.2).

Moreover, we provide the code of three major components of MuHS in the supplementary and will release the official implementation upon acceptance.

---

### Decision · Program_Chairs · 2023-01-20

**Decision:**

Accept: poster

**Justification For Why Not Higher Score:**

Although the proposed method is solid and demonstrates effective performance, it does not provide significantly novel ideas to justify a higher score.

**Justification For Why Not Lower Score:**

I do not recommend reject, since this paper presents a reasonable method for a useful problem, and provides solid experimental validation that was improved with an additional dataset during the author rebuttal period.


**Metareview: Summary, Strengths And Weaknesses:**

This paper initially received borderline scores, and was therefore additionally discussed in a virtual meeting with reviewers. Reviewers generally considered the paper to be solid, and that it proposed a reasonable approach to a useful problem, although there were some comments that the method is somewhat incremental. There were also initially some concerns about experimental evaluation and testing on too few datasets, however the author rebuttal added results on an additional dataset and overall addressed most reviewers concerns. Some reviewers raise their scores in light of the rebuttal. At the reviewer meeting, all reviewers present were in favor of accepting the paper. They only reviewer not present was also the only one with a score of marginally below the acceptance threshold; however, one of their main comments requested more datasets which the remaining reviewers and I felt was addressed with the author rebuttal. In summary, I agree with the majority reviewer opinion that this paper should be accepted to the conference.

**Note From Pc:**

if the above contains the word "oral" or "spotlight" please see: "oral" presentation means -> notable-top-5% and "spotlight" means -> notable-top-25%. As stated in our emails, we are disassociating presentation type from AC recommendations

**Summary Of Ac-Reviewer Meeting:**

At the meeting, it was discussed that the author rebuttal addressed most reviewer concerns, especially with respect to experimental validation with the addition of another dataset. Given that there were no major concerns remaining, an accept is reasonable. Only one reviewer out of 4 did not show up, despite multiple reminders, and had not responded to the author rebuttal, but the rest of the reviewers and I felt that their major concerns were addressed by the author rebuttal.